# Hepatoprotective Activity of Lignin-Derived Polyphenols Dereplicated Using High-Resolution Mass Spectrometry, In Vivo Experiments, and Deep Learning

**DOI:** 10.3390/ijms232416025

**Published:** 2022-12-16

**Authors:** Alexey Orlov, Savva Semenov, Gleb Rukhovich, Anastasia Sarycheva, Oxana Kovaleva, Alexander Semenov, Elena Ermakova, Ekaterina Gubareva, Anna E. Bugrova, Alexey Kononikhin, Elena I. Fedoros, Evgeny Nikolaev, Alexander Zherebker

**Affiliations:** 1Skolkovo Institute of Science and Technology, 30 Bolshoy Boulevard bld. 1, 121205 Moscow, Russia; 2Moscow Institute of Physics and Technology, 141701 Moscow, Russia; 3N.N. Petrov National Medical Research Center of Oncology, 68 Leningradskaya Str., 197758 Saint Petersburg, Russia; 4Emanuel Institute for Biochemical Physics, Russian Academy of Sciences, 119334 Moscow, Russia

**Keywords:** NAFLD, hepatoprotective activity, complex mixtures, polyphenolic composition, dereplication, labeling, mass spectrometry, structure

## Abstract

Chronic liver diseases affect more than 1 billion people worldwide and represent one of the main public health issues. Nonalcoholic fatty liver disease (NAFLD) accounts for the majority of mortal cases, while there is no currently approved therapeutics for its treatment. One of the prospective approaches to NAFLD therapy is to use a mixture of natural compounds. They showed effectiveness in alleviating NAFLD-related conditions including steatosis, fibrosis, etc. However, understanding the mechanism of action of such mixtures is important for their rational application. In this work, we propose a new dereplication workflow for deciphering the mechanism of action of the lignin-derived natural compound mixture. The workflow combines the analysis of molecular components with high-resolution mass spectrometry, selective chemical tagging and deuterium labeling, liver tissue penetration examination, assessment of biological activity in vitro, and computational chemistry tools used to generate putative structural candidates. Molecular docking was used to propose the potential mechanism of action of these structures, which was assessed by a proteomic experiment.

## 1. Introduction

According to a recent analysis [1], acute and chronic liver diseases remain a significant public health threat in Europe with more than 1 billion affected people, with increasing incidence and prevalence of these diseases in industrialized countries [1,2,3,4]. Among the most common liver diseases worldwide is the non-alcoholic fatty liver disease (NAFLD). NAFLD is the leading cause of liver-related morbidity and mortality, [3] with more than 800 million people suffering from this disease. Therefore, an urgent task is the search and development of new hepatoprotective drugs targeting NAFLD with various mechanisms of action [5,6].

Natural and synthetic complex mixtures of low-molecular-weight polyphenolic compounds, including products of lignin oxidation, possess a wide range of biological activity and low toxicity, which makes them promising candidates for therapeutic applications [7,8]. It has been shown that polyphenols can act as traps for free radicals [8,9] and therefore prevent oxidative stress [10]. In addition to that, they possess an anti-inflammatory effect, which is manifested in the suppression of the synthesis of cytokines and interleukins of the pro-inflammatory spectrum [11]. Depending on the origin, natural mixtures can exhibit hepatoprotective properties, including the aforementioned prevention of oxidative stress (e.g., silibinin, catergen), stimulation of liver parenchyma regeneration [12], and alleviation of NADFL biochemical symptoms [13,14]. However, most of the papers published on the biological activity of complex mixtures are related to recording the biological response in experiments in vitro and in vivo, while their molecular mechanism of action remains unknown. The main reason is the high complexity of the analysis of multi-component systems at the molecular level and the lack of feasible opportunities to identify components responsible for their action.

The molecular complexity of nature-derived mixtures is usually analyzed in the framework of the dereplication concept [15]. The dereplication procedure is aimed at the quick identification of known chemotypes in the mixture by analyzing the spectra data and searching for components in biological activity and spectra databases. Dereplication strategies are based on the combination of advanced analytical methods including HPLC, MS, NMR, biological tests, etc., complemented with data mining techniques [15,16] Structural information is used to narrow a list of possible candidates from open-source libraries such as ChEMBL [17] and CoCoNut [18]. Yet, the combinatorial nature of highly degraded polyphenols prevents them from obtaining sufficiently narrow fractions [19]. Additionally, the application of tandem mass spectrometry within the dereplication strategy is almost impossible due to its time-consuming nature, even with the most powerful Fourier transform ion cyclotron resonance mass spectrometry (FTICR MS) techniques [20]. Instead, the integral structural parameters are investigated in order to build up a chemical foundation for biological properties [21,22]. In addition, the correlation of the abundance of molecular components with the biological response in a set of similar samples or during bulk fractionation can be used to prioritize components in a mixture [23,24]. Ultimately, this approach mostly misses structural information, preventing it from discovering the mechanism of action.

Alternatively, the administration of labeled mixtures facilitates the detection of components in tissues. For example, in previous reports, the application of tritium labeling allowed authors to quantitatively analyze tissue distribution and pharmacokinetic parameters for synthetic polyphenols administrated to mice [25]. The drawback of the approach is the total loss of qualitative information. In this study, we aimed to adapt a labeling strategy in order to detect individual components of the complex oxidized lignin derivative BP-Cx-1 [26] in mice liver tissue by FTICR MS as well as to explore their structural features by application of selective chemical tagging. Molecular and structural information obtained with these methods was used to apply deep learning to generate candidate structures for the detected components, suggest a molecular mechanism of hepatoprotective activity, and, finally, extract a more active fraction enriched with these components. Based on the data obtained in silico, an in vivo experiment on the evaluation of the hepatoprotective activity of BP-Cx-1 and its fraction was carried out in a model of NAFLD in mice with induced type 2 diabetes.

## 2. Results

### 2.1. Overview of the Study Pipeline

To assess which chemotypes can potentially lead to bioactivity, we applied the following computational methodology (Figure 1). In the first step, the molecular composition of a complex polyphenolic mixture was characterized using FTICR MS, and its bioactivity and the potential active components were elucidated by the extraction of D-labeled material from mice livers followed by MS analysis. Secondly, bulk fractionation of the parent material was conducted to obtain a fraction enriched with the detected components. Its hepatoprotective activity was compared to the parent material. Then, to indicate a potential mechanism of action, a combination of selective chemical labeling, FTICR MS, proteomics, and computational chemistry methods was applied. The Kelch-like ECH-associated protein 1 (KEAP1)–nuclear factor erythroid 2–related factor 2 (Nrf2) pathway, which plays an important role in NADFL, was chosen as a primary target for analyzing the chemical space of potential components. The ability of the active fraction to interact with KEAP1 was evaluated by LC-MS experiments, and the putative structural candidates were suggested using deep learning and molecular modeling.

### 2.2. Determination of BP-Cx-1 Components in Mice Liver

BP-Cx-1 was analyzed by FTICR MS before and after it was labeled. The labeling method was effective—most of the components were multiply labeled (Appendix A, Appendix A). Despite NaOD provided for the carbon-skeleton labeling, back-exchange during metabolism might occur and deep labeling assured detection of components of interest after administration, tissue sampling, and extraction described in Methods. The set of extracts included control samples, samples after the administration of parent BP-Cx-1, and samples obtained after D-labeled BP-Cx-1 administration. As it is described in Methods, only the latter deuterium-contained samples were used for component detection. FTICR mass spectra were highly populated (Appendix A) and visually highly resembled, which indicated low intensities of BP-Cx-1-derived components. Still, applying the developed spectra-treating algorithm, 448 and 312 D-labeled were found in the liver extract after parenteral and oral administration, respectively, comprising 644 unique records. It should be noted that a number of these molecules were missing in the mass spectra of D-labeled BP-Cx-1. This might be explained by the reproducibility issue of direct-ESI measurements, in which charge suppression, combating ionization, and even the lifetimes of ion clouds depend on the mixture composition. After additional filtration of the data, the list of 208 molecular components of BP-Cx-1 was composed presenting formulae observed either after parenteral or oral administration. For structural characterization of BP-Cx-1 components detected in the liver, selective deuteromethylation, reducing, and mild bromination reactions were conducted giving reliable information on the number of carboxylic and carbonyl groups, as well as aromatic rings. Upon aggregating 133 out of 208 formulae, all three descriptors were obtained (Appendix A). The distribution of these moieties over the van Krevelen diagram (Appendix A) shows that most of the BP-Cx-1 component contained at least one carbonyl group and most of the compounds contained two carboxylic groups, and they were mostly aromatic.

### 2.3. Fractionation of BP-Cx-1 for Enrichment of Targeted Component

We postulated that components found in the liver should be responsible for the hepatoprotective properties of BP-Cx-1. Therefore, it was promising to isolate the fraction enriched with these components. To do so, we performed solvent extraction in the Soxhlet apparatus with a gradual increase in binary solvent polarity: From 100% CH_3_OH to 25% CH_3_OH-75% H_2_O. All fractions were analyzed by FTICR MS. The 100% CH_3_OH and 25% CH_3_OH fractions were composed of the maximum and minimum numbers of formulae: 5927 and 3676, respectively. Number-averaged values obtained from mass-spectra were similar; all fractions were lacking oxidized compounds—(O/C)_n_ values were below 0.3. The advantage of FTICR MS is that we can examine groups of compounds, which contribute to the number-average values. This is important since the presence of impurities (e.g., lipids) may affect the average values significantly. Correspondingly, we grouped compounds according to their atomic ratios and constrained aromaticity index excluding lipids, which are unlikely indigenous to the polyphenolic material (Appendix A). Here, it is clearly seen that, in fact, the gradient in the molecular composition is pronounced. It is obvious that the gradient extraction resulted in the shift from mostly unsaturated compounds to aromatic and, consequently, to condensed compounds dominating 50% and 25% CH_3_OH fractions. Schematically, this trend is designated on the van Krevelen diagram (molecular profiles) (Figure 2A). It is worth noting that for natural samples, such as humic substances, the shift in elemental composition during fractionation is usually explicit. This indicated high structural diversity of molecular constituents of BP-Cx-1 on the relatively narrow range of elemental compositions (in terms of O/C and H/C ratios), which corroborates well with the distribution of structural descriptors on van Krevelen diagrams (Appendix A). All fractions were examined as present and contributed to the total intensity of 208 molecular components, which were found in the liver in the previous step. The methanol fraction was the most enriched of those components (Figure 2B). At the next stage, the biological activity of parent BP-Cx-1 and the presumably most active fraction were examined.

### 2.4. Assessment of Biological Activity of Parent and Fractionated BP-Cx-1

General information on body weight, food and liquid consumption, and blood analysis can be found in Appendix A. During the first two weeks, the weight of animals from groups with induced diabetes—negative control (group 2), under parental BP-Cx-1 treatment (group 3), and under BP-Cx-1 fraction treatment (group 4)—was comparable to that of the intact control (group 1). After the administration of streptozotocin (week 3), a decrease in body weight was observed for experimental groups 2–4 with no influence of the treatment. It was accompanied by a decrease in food consumption and an increase in water consumption. By the end of the experiment, the increase in hematocrit was observed for group 2. The treatment with both BP-Cx-1 and its fraction normalized the hematocrit level (Appendix A). Unlike a routine blood test, blood chemistry analysis showed a significant deviation of many parameters (Table 1). Compared to group 1, all experimental groups showed an increase in the total protein level, mostly due to the increase in globulin content. The total protein level increased up to 61 g/L after the administration of streptozotocin, which may indicate inflammation. The treatment with an isolated fraction (group 4) almost normalized the total protein (~47 g/L) and significantly decreased the globulins content (down to ~24 g/L). All experimental groups showed a significant increase in liver enzymes indicating damage to liver functioning. The administration of BP-Cx-1 and its active fraction did not result in a significant decrease in AST and ALT levels. A similar trend was observed for glucose. The positive effect of the treatment was observed for pigment metabolism. Liver steatosis in group 2 was accompanied by increased bilirubin levels—up to ~18 µmol/L, in which 93% was accounted for by the conjugated bilirubin. The administration of fractionated BP-Cx-1 (group 4) decreased the total bilirubin level to ~15 µmol/L and conjugated bilirubin to ~13 µmol/L. Bilirubin is currently considered an endogenous antioxidant cytoprotectant [27] so its level is likely regulated by the administration of the exogenous polyphenol agent. This corroborates well with Group 3 (parent BP-Cx-1) with bilirubin levels comparable to group 2. Lipid exchange was also affected by inflammation. The cholesterol level increased in group 2 up to 3.5 µmol/L. Similar to bilirubin, only the isolated fraction of BP-Cx-1 (group 4) decreased the cholesterol level to 2.83 µmol/L. Despite the significant deviation of experimental groups from group 1, the positive therapeutic effect of the isolated fraction was pronounced. This supports the assumption of a leading role of liver-isolated components in the overall hepatoprotective effect of BP-Cx-1. In the next step, their molecular mechanism of action is studied.

### 2.5. Examination of BP-Cx-1 Interaction with KEAP1

The commercial KEAP1 protein was investigated using the conventional bottom-up proteomic approach and its sequence coverage was substantial (~40%) with post-translation modifications (PTM) resolved. In the discovered sequence, the electrophilic inhibitor could modify eleven cysteine residues, including Cys273, which, according to the literature, plays a major role in the KEAP1–Nrf2 interaction during oxidative stress. The challenge in the analysis of the BP-Cx-1 interaction with KEAP1 lies in the ambiguity of the mechanism and the unknown structure of any of the 208 liver-detected components. Consequently, the protein–ligand interaction was examined by the inhibition of KEAP1 alkylation and the comparison of relative intensities of the free-cysteine-containing peptides with and without incubation with BP-Cx-1. The results are presented in Figure 3.

It can be seen that after incubation, the relative intensities of cysteine-alkylated peptides varied from the control sample. The relative intensities significantly decreased for Cys226, Cys241, Cys249, and Cys273 as compared to other fragments. Changes in the relative intensities of alkylated peptides are the result of KEAP1’s interaction with BP-Cx-1. This interaction was not selective but there was clear targeting. It is likely that the environment of the interactive cysteine targeted BP-Cx-1 in these sites. It is particularly important that the Cys273 was one of the reactive sites and was a signal residue. Upon obtaining information about the biological activity of BP-Cx-1 both in vitro and in vivo, its possible molecular mechanism was further investigated using computational tools.

### 2.6. Generation of Structural Candidates for BP-Cx-1 Active Components Using Deep Learning

In this work, the deep learning method [28] based on the recurrent neural network architecture (RNN) was used for the generation of natural-like compounds structurally similar to compounds active against KEAP according to ChEMBL data (see Section 4). In total, 1.8 M unique valid SMILES strings were generated. The generated compounds comprise a wide range of various natural product-like scaffolds similar to those present in CoCoNut (Figure 4a) and the compounds interfering with the Nrf2 pathway (Figure 4b, Appendix A). The distribution of occurrence of physico-chemical descriptors (Appendix A) among the generated compounds and compounds from CoCoNut and Nrf2 subsets is similar. Matching of the generated compounds to formulae obtained with FTICR MS and structural features of compounds determined the H/D exchange significantly reduced the number of putative structural candidates (1775 compounds). The compounds were then used to investigate their ability to interact with KEAP1 using molecular docking.

### 2.7. Molecular Docking of Structural Candidates to KEAP-1 Structure

While the KEAP1–Nrf2 interaction is widely acknowledged as an important target for the design of new therapeutics [29] for various chronic diseases, the structural biology of KEAP1 is not completely known. KEAP1 is a 624-amino-acid protein, which consists of five domains (Figure 5a). The X-ray structures in the apo form and with ligands are available only for the Kelch-like domain (e.g., PDB ID 6LRZ [30]) and BTB domains (e.g., PDB ID 4CXI [31]).

With the recent advances in predicting the 3D protein structures using deep learning [32] and the deposition of the large pool of predicted structures in the AlphaFold Protein Structure Database [33], a new avenue for structural biology studies has been paved. We retrieved the structure of the full-length (624 amino acids) KEAP1 predicted by AlphaFold (Figure 5a) and used it to investigate the ability of the generated compounds to interact with the protein. Unsurprisingly, Kelch-like and BTB domains of the model were structurally similar to the aforementioned available X-ray data (RMSD < 1 A for Cα atoms) differing mostly in loops’ conformation. Considering that the cysteine residues in human KEAP1 including Cys 151, 273, and 288 are highly reactive and play an essential role in the repression of Nrf2 activation, we addressed the question of whether the molecular components of the sample can prevent the binding of the electrophiles to these residues [34]. Proteomics experiments (see above) did not reveal signatures of covalent binding but rather allowed us to identify cysteines affected by the presence of the sample. Therefore, we hypothesized that putative compounds could bind to the protein in close proximity to these residues. In particular, we focused our attention on Cys273, since this residue is buried in the potentially “druggable” cavity between helixes formed by 298–311 and 262–275 residues. The binding of the compounds to this site can barricade access to Cys273 and thus prevent the alkylation of the cysteine moiety. Molecular docking simulations revealed that the generated compounds can indeed bind to this cavity (Figure 5b and Appendix A), anchoring to the site via the π–π stacking interaction with His274. The presence of positively charged amino acid residues framing the entrance to the site is in line with the mostly nucleophilic nature of the molecular components of the sample: Many components contain at least one −OH or −COOH group (−COO− at the physiological pH).

**Figure 5 ijms-23-16025-f005:**
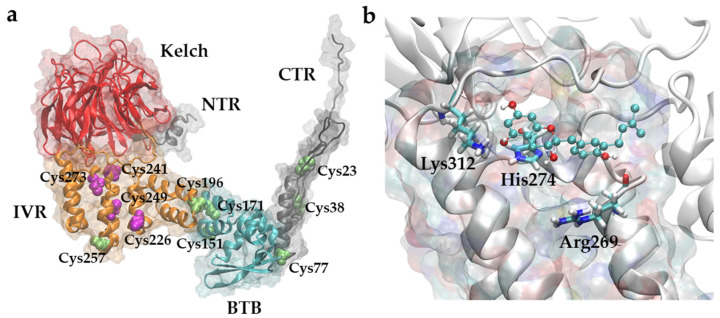
(**a**) A model of full-length KEAP1 predicted by AlphaFold2. A BTB domain (amino acids 61–179) is shown in cyan, an intervening region (IVR, amino acids 180–314) is shown in orange, and a Kelch domain (Kelch) is shown in red. N-terminal (NTR, amino acids 1–60) and C-terminal regions (CTR, amino acids 599–624) are shown in grey. Cysteine residues, which were investigated in the proteomics experiments, are shown in magenta (non-alkylated ones) and lime (alkylated). Domains are designated as in ref. [35]. (**b**) Docking pose of the generated compound (3,5,7-trihydroxy-2-[(4-hydroxy-3-(3-methylbut-2-en-1-yl)phenyl)methyl]-2,3-dihydro-4H-1-benzopyran-4-one), shown as ball-and-stick model) in the region close to Cys273. Positively charged amino acids (Lys312, His274, Arg269) forming the entrance to the cavity are shown. Visualization is made in VMD 1.9.3. [36].

## 3. Discussion

Numerous biological pathways are associated with NAFLD [37,38], and therefore, various molecular targets can be exploited for the design of NAFLD-targeting therapeutics [5]. While a thorough analysis of a sample’s mechanism of action assumes the analysis of its interaction with a panel of known molecular targets associated with NAFLD, such an analysis was outside the scope of this paper. Alternatively, in this paper, we suggest an example of a dereplication pipeline combining computational and experimental methods to decipher putative chemotypes behind the activity of the complex mixture under study. The research design presented in Figure 1 is advantageous in the case of a multicomponent mixture. In fact, many structures present in the lignin derivative may interact with KEAP1 or other NAFLD-related proteins. However, for future development, it is important to determine which component may penetrate the liver tissue in detectable amounts.

This was achieved in our investigation by the administration of D-labeled material. Components of BP-Cx-1 are characterized by structural ambiguity; consequently, it is a non-trivial task to detect them in tissue. The application of the H/D exchange facilitated the determination of BP-Cx-1 in the extract due to the low abundance of heavy isotopes in the organism. Thus, labeled (exogenous) and endogenous molecules were easily distinguished by mass spectrometry. We suggest that detected components are responsible for hepatoprotective activity. In support of this, a fraction enriched with these components demonstrated a higher protective effect compared to the parent material.

The major focus of the research was chemoinformatics approaches for structure elucidation in order to assess the molecular mechanism. While the mining of candidate structures in large repositories of chemical data can be an effective approach to dereplication [23,26], the number of compounds with biological activity assessed experimentally is limited, which leads to small coverage of potential chemotypes for various formulae. This problem was addressed by using methods of de novo structure generation [39]. These approaches allow the rapid generation of millions of compounds bearing chemical features and/or possessing (predicted) required bioactivity profiles. The generated structures did not exhaustively cover the chemical space of relevant structural isomers and were considered “probes” of chemotypes that are natural-like and similar to the compounds interfering with the Nrf2 pathway. Yet, 1.8 M structures were generated. In order to obtain a feasible list of candidates, we narrowed the structure space by the results of selective chemical tagging. This yielded 1775 structures, which were possible to handle without excessive calculation power. Indeed, molecular docking enabled us to suggest a binding site for these structures, which was supported by a proteomic experiment with KEAP1. Moreover, structural information on the BP-Cx-1 components suggests a blocking mechanism for Cys273 residue resulting from the π–π stacking interaction with adjacent His274. However, it should be emphasized that numerous sites on the surface of the KEAP1 model can be determined (Appendix A), and it is known that compounds can bind to several sites at once [30]. Therefore, further experimental investigation of mixture components’ binding modes via X-ray crystallography is required.

Overall, this pilot study provides a new strategy to study the mechanism of action of natural multicomponent mixtures. This requires a combination of chemical, chemoinformatic, and biological experiments. In the particular case of BP-Cx-1, the results are promising from a practical point of view. In fact, synthetic lignin derivatives have batch-to-batch reproducibility [40]. With the proven mechanism, it may be introduced as a bivalent drug: As a dietary fiber, which helps in the elimination of residues in the intestinal tract and improves the intestinal environment [41,42], and as a hepatoprotective agent, which prevents or reduces liver damage.

## 4. Materials and Methods

Solvents and other reagents used in this study were commercially available. Methanol of HPLC grade (Chimmed, Moscow, Russia) was used for the elution and dissolution of samples. High-purity distilled water (18.2 MΩ) was prepared using a Millipore Simplicity 185 system (Laverna XX1, Moscow, Russia). Bond Elut PPL (Priority PolLutant, Agilent Technologies, Santa Clara, CA, USA) cartridges (50 mg, 1 mL) were used for the isolation and purification of the parent and labeled samples. Raw mass spectrometric data for deuteromethylation and reducing reactions were obtained from a previous study [43]. H/D exchange of the parent BP-Cx-1 was performed according to the previously described procedure [44]. For the bromination of BP-Cx-1, 5-fold excess of NBS (Chimmed, Moscow, Russia) was added to the AcN (Chimmed, Moscow, Russia) solution and stirred overnight followed by PPL extraction.

### 4.1. Enumeration of Functional Groups and H/D Exchanges

Detailed information on FTICR MS measurements can be found elsewhere [45]. For the enumeration of COOH− and C=O groups, as well as the presence of an aromatic ring, a Python-based script was developed, which enabled fully automatic assignment [46]. In brief, the algorithm facilitates the extraction and enumeration of peak series with mass differences corresponding to the labeling reaction and filtration of the obtained results based on the heuristic chemistry-based rules depending on the reaction. Mass differences were 17.03448, 3.02193, 77.91051, and 1.00628 Da for singly charged ions corresponding to deuteromethylation, reducing, bromination, and H/D exchange. In all cases, the mass error (between peaks of a labeling series) was below 0.0003 m/z, which was optimized based on the performance of the FTICR MS instrument.

### 4.2. Fractionation of BP-Cx-1

Fractionation was performed by using different combinations of binary solvent mixtures from 100% CH_3_OH to 25% CH_3_OH-75% H_2_O in the Soxhlet extractor. For this, 50 g of the solid BP-Cx-1 batch was used. All fractions were analyzed by FTICR MS without further purification. The fraction of interest (100% CH_3_OH) was dried, yielding 5 g (105) of the solid material.

### 4.3. Animal Welfare

Female BALB/c mice (animal facility Stolbovaya, Moscow District, Russia) were quarantined for 14 days upon delivery. The animals were kept in T2 type IVC cages under artificial 12 h light/dark cycle conditions, 21 ± 2 °C, average humidity of 20–50%, and ad libitum access to laboratory chow (Laboratorkorm LLC, Moscow, Russia) and tap water. Experimental animals were handled under the Guide for the Care and Use of Laboratory Animals, 8th edition. The study protocol was approved by the Local Ethics Committee of the N.N. Petrov National Medical Research Center of Oncology (protocol no. 3/215; dated 22 September 2020). Experimental animals at the end of observation were euthanized by decapitation under isoflurane anesthesia.

### 4.4. Induction of NAFLD Accompanied by Type 2 Diabetes

NAFLD accompanied by type 2 diabetes was induced as follows [47,48]: Throughout the experiment (7 weeks), BALB/c female mice were put under a high-carbohydrate diet (5% aqueous fructose solution ad libitum) and a high-fat diet (standard chow + lard). Pancreatic lesions were induced via intraperitoneal streptozotocin (CarboSynth, San Diego, CA, USA) at a dose of 80 mg/kg (week 3). Seven days after this, blood glucose levels were measured in the animals. Animals were included in experimental groups provided their glucose concentrations were above 10 mmol/L (180 mg/dL)).

After randomization by glucose level, four animal groups of 8 mice each were formed: The intact control (group1), mice with induced type 2 diabetes and pancreatic lesions without treatment—negative control (group2), and mice under treatment of the parent BP-Cx-1 (group 3) and its methanol fraction (group 4). Both parent BP-Cx-1 and its fraction were administrated at a dose of 75 mg/kg body weight daily by gavage. Blood and serum samples were collected after euthanasia at the end of the experiment.

### 4.5. Extraction of BP-Cx-1 Components from Mice Liver Tissue

For the pharmacokinetic experiment, 29 Balb/c female mice were divided into 5 groups: Control (3), intravenous administration of parent and labeled BP-Cx-1 (3 and 10, respectively), and gavage administration (3 and 10, respectively). Intravenous and gavage doses were 20 µg/kg and 100 µg/kg, respectively. Intravenous administration was carried out using a plastic restrainer without anesthesia. All animals underwent euthanasia after 2 h following the maximum concentration of the material in tissue obtained previously by tritium labeling [25]. The organs were collected, and the liver was thoroughly cleaned of surrounding tissues and washed of blood in a cold solution of 0.9% sodium chloride (400 mL, the solution was changed after every 3 animals at one time point). Then, fluid residues were removed using filter paper, and livers were frozen for further extraction.

Before extraction, liver tissues were ground in a mortar in liquid nitrogen pooled according to the mice grouping and freeze-dried. First, lipids were removed by three-fold washing with a chloroform-methanol (2:1) solution. The solid residue was dried again and extracted via water with continuous shaking (3 h) followed by chloroform extraction to remove polar metabolites. The water residue was extracted by solid-phase extraction on PPL resin in two steps. First, the supernatant was passed at a native pH through the first cartridge to remove metabolites. The collected permeate was acidified to pH 2 and passed through the second cartridge targeting BP-Cx-1 components. Components were further eluted with methanol and kept in the freezer for FTICR MS analysis.

### 4.6. BP-Cx-1-KEAP Interaction

For the examination of the interaction between the BP-Cxp1 active fraction and KEAP-1, the conventional proteomic bottom-up approach was applied with and without overnight incubation of the protein with 0.6 mg/L of polyphenol material in Tris buffer (50 mM, pH 8). To both samples, dithiothreitol (DTT) was added as a reducing agent, and the solution was incubated for 1 h at 37 °C for the reduction of disulfide bonds in the KEAP-1 protein. Further, acetylation with iodoacetamide was conducted. Before trypsin digestion, 0.01% detergent was added to facilitate better protein coverage. Digestion was quenched by the addition of 1% FA. The tryptic peptides were analyzed in triplicate on a nano-HPLC Dionex Ultimate 3000 system (Thermo Fisher Scientific, Madison, WI, USA) coupled to a timsTOF Pro (Bruker Daltonics, Billerica, MA, USA) mass spectrometer. The sample volume was 2 µL per injection. HPLC separation was carried out using a packed emitter column (C18, 25 cm × 75 µm 1.6 µm) (Ion Optics, Parkville, Australia) by gradient elution. Mobile phase A was 0.1% formic acid in water; mobile phase B was 0.1% formic acid in acetonitrile. LC separation was achieved at a flow of 400 nL/min using a 40 min gradient from 4% to 90% of phase B. The obtained data were analyzed using PEAKS XPro software (BSI, North Waterloo, ON, Canada) using the following parameters: Parent mass error tolerance of 20 ppm and fragment mass error tolerance of 0.03 Da. Only the peptides with both trypsin-specific ends were considered for identification purposes. The alkylation of cysteine residues was set as a possible variable modification.

### 4.7. ChEMBL Bioactivity Data Mining

The MySQL version of ChEMBL (v. 29) [49,50] was retrieved and installed on a local MySQL server. The Python library “mysql-connector-python” (8.0.18) [51] was used to access data from the MySQL server. Data retrieval, standardization, and preprocessing were performed using pandas [52,53] and RDKit (v. 2019.03.3) [54] libraries in the Jupyter notebook interactive environment.

Assays with various types of activity data against Nrf2 were retrieved using a list of key substrings (“%nrf2%”, “%nfe212%”, “%nuclear%factor%erythroid%2%related%factor%2%”) from target_dictionary.pref_name, assays.description, assay.organism, and target_dictionary.organism fields. Retrieved assay entries were manually checked. Only compounds consisting of carbon, hydrogen, and oxygen were kept. Activity entries were standardized, and 1/0 (“active/inactive”) labels were assigned to each entry (see GitHub repository). Structures were standardized following the protocol from https://github.com/PharminfoVienna/Chemical-Structure-Standardisation (accessed on 8 Septmber 2021) with slight modifications. Stereochemistry was depleted from SMILES entries. The activity entries related to the same standardized SMILES were grouped, and the mean values were taken. Compounds with average activity values no less than 0.5 were designated as “active”, and related SMILES notations were retrieved. Finally, the dataset composed of 1942 smiles entries was used for tuning (see below). The complete protocols used for data retrieval and standardization are available in the Jupyter notebook in the GitHub repository (see below).

### 4.8. Generation of Structural Candidates Using Deep Learning

#### 4.8.1. Generation of Compounds

The recurrent neural network with the architecture described by Segler, M.H.S. et al. [28] was used to generate putative structural candidates. The hyperparameters of the original network were used. At first, a model was trained on an entire CoCoNut dataset for 40 epochs. The model was then retrained on the dataset comprising compounds active against Nrf2 for 10 epochs. This fine-tuned model was used for sampling.

#### 4.8.2. Filtering of Generated Compounds Using Labeling Data

Generated SMILES were checked for validity using RDKit, and duplicates were removed. Corresponding compounds were filtered according to the data from isotopic labeling experiments: The compound should have contained at least one ionizable group (-OH, -COOH) and its functional groups should have matched at least one out of three enumerated moieties: Aromatic, carboxylic, and carbonyl. The counts of relevant functional groups were calculated using RDKit.

#### 4.8.3. Chemical Space Analysis

All procedures related to the analysis of chemical structure datasets were performed using either RDKit or DataWarrior v.5.5.0 [55]. Self-organizing maps were built in DataWarrior using FragFP fingerprints. Scaffold analysis was performed in DataWarrior.

### 4.9. Molecular Docking of Structural Candidates to KEAP-1 Structure

#### 4.9.1. Preparation of Receptor

The model of the full-length KEAP-1 structure generated by AlphaFold2 (UniProt KB [56] identification code P57790) was retrieved from https://alphafold.ebi.ac.uk/ (accessed on 1 October 2021). The structure was minimized using the OpenMM [57] program via python script. The following parameters were used: A step size of the Verlet Integrator of 0.001 ps, 1000 steps of minimization, and an AMBER99SB force field. The quality of the minimized structure was checked using PROCHECK [58] and Prosa-Web (Appendix A) [59,60]. The parameters of the box for docking selected to be close to the Cys273 residue were chosen using AutoDock Tools [61]: The coordinates (−5, −18, −5) and dimensions of 20 × 20 × 20 A°.

#### 4.9.2. Binding Site Detection

Binding site detection was performed using fpocket2 [62]. The default parameters were used.

#### 4.9.3. Preparation of Ligands

Initial conformations of ligands were obtained using Open Babel 2.4.1. [63] The following options were used: gen3d, best (for more thorough conformation optimization), and -p 7.4 (adding hydrogens appropriate for pH 7.4). Tautomers and stereoisomers were not enumerated, and the default tautomer/stereoisomer generated by Open Babel was kept.

#### 4.9.4. Molecular Docking

Molecular docking was performed using AutoDock Vina v.1.2.2. [64,65] with the exhaustiveness parameter equal to 32. All other parameters were left as default.

#### 4.9.5. Visualization and Analysis

For visual analysis, VMD 1.9.3 [36] and YASARA [66] were used. Post-docking analysis was performed in python.

## 5. Conclusions

The investigation of the synthetic multicomponent polyphenolic agent, BP-Cx-1, showed its pronounced hepatoprotective activity in mice. The developed H/D exchange labeling approach enabled us to find 200 individual components in mice livers after its administration. This approach seems promising, evidenced by the increased activity of the isolated fraction, enriched with the detected components. The H/D exchange was analytically justified and maintained the properties of BP-Cx-1. The additional combination of labeling techniques, data mining, and deep learning enabled us to generate 1775 natural-like structures with potential hepatoprotective activity and molecular formulae corresponding to components found in the liver. Further application of molecular docking showed the possibility of interaction of the chosen structures with positively charged amino acid residues framing the cavity bearing Cys273. Further experimental investigation into mixture components’ binding modes via X-ray crystallography is required to prove this prediction.

## Figures and Tables

**Figure 1 ijms-23-16025-f001:**
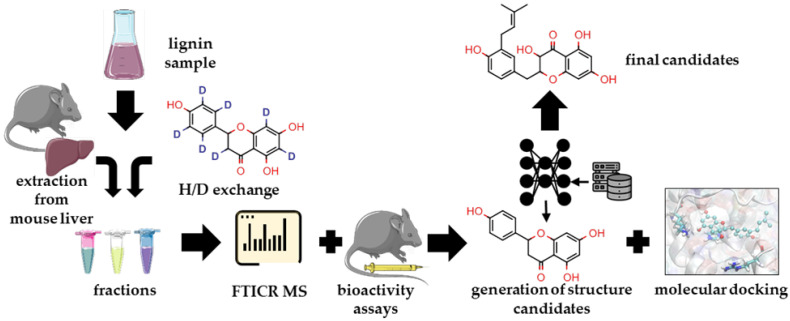
The dereplication pipeline used in this work. H/D exchange was used to label lignin-derivative and find its component in liver by FTICR MS. The sample and its enriched fraction were tested on its ability to alleviate NADFL-related biochemical changes. LC-MS experiments assessing the ability of samples to interact with the KEAP-1 were then performed. Further, a recurrent neural network was trained on the natural compound structures from CoCoNut and fine-tuned on compounds possessing activity against KEAP1-Nrf2 system retrieved from ChEMBL. The network was used to sample various natural compound-like structures that were filtered by the relevance according to the experimental MS data including isotopic labeling. Finally, the ability of the filtered compounds to interact with the KEAP-1 was assessed using molecular docking simulations.

**Figure 2 ijms-23-16025-f002:**
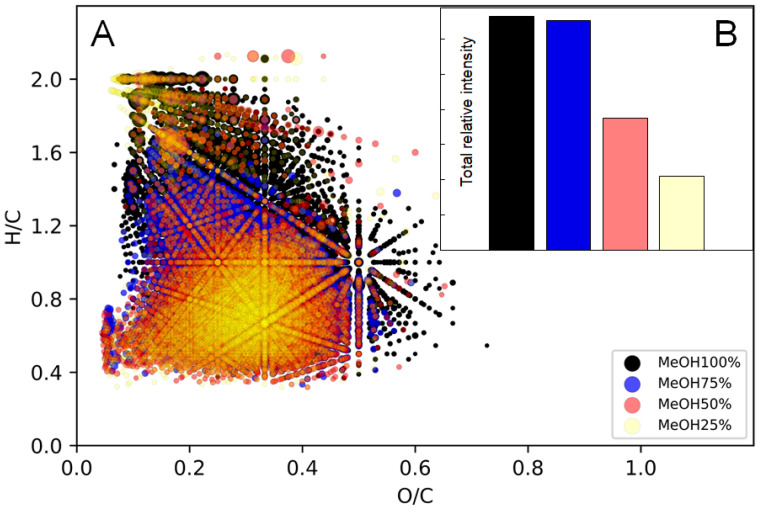
Van Krevelen diagrams for BP-Cx-1 fractions obtained by gradient extraction (**A**); summarized contribution of molecular components found in liver to the total intensity in mass-spectra of fractions (**B**).

**Figure 3 ijms-23-16025-f003:**
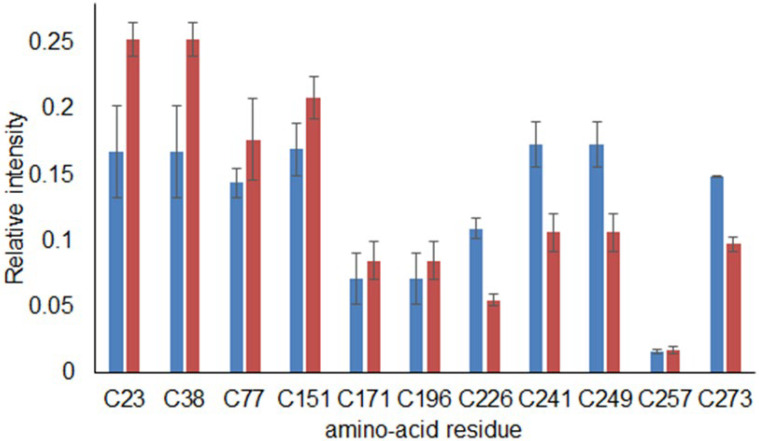
Relative intensities of alkylated peptides containing free cysteine without (blue) and with (red) incubation with BP-Cx-1 determined by LC-MS.

**Figure 4 ijms-23-16025-f004:**
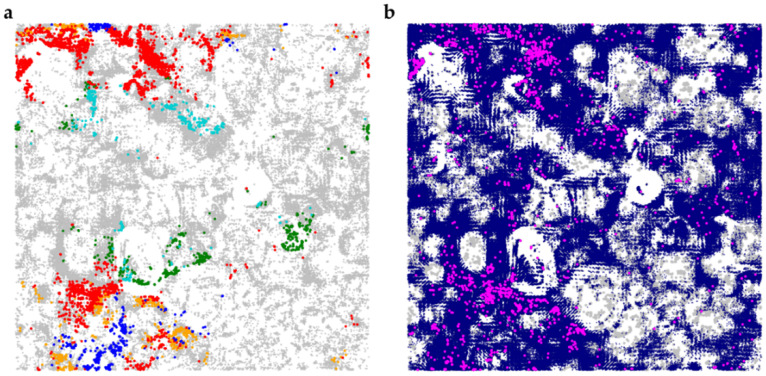
Comparison of the chemical spaces of natural products from CoCoNut database (only CxHyOz), compounds active against Nrf2 and generated compounds using self-organizing map (SOM) (100 × 100 neurons, FragFp fingerprints, see Materials and Methods for details). (**a**) The SOM built on CoCoNut compounds colored by most frequent scaffolds (see Appendix A). Red—benzene ring, cyan—sterone ring, orange—coumarin, blue—flavone, green—tetrahydropyran, grey—compounds bearing other scaffolds. Markers for compounds bearing frequent scaffolds were enlarged for better visibility. (**b**) The SOM built on CoCoNut compounds (grey) on which generated compounds (dark blue) and compounds active against Nrf2 (magenta) were projected. Markers for CoCoNut compounds and compounds active against Nrf2 were enlarged for better visibility.

**Table 1 ijms-23-16025-t001:** Biochemical blood test of experimental mice.

Group	Total Protein, g/L	Albumin, g/L	Globulins, g/L	Amylase, U/L	AST, U/L	ALT, U/L
1	44.4 ± 0.8	28.0 ± 0.47	16.4 ± 0.90	664 ± 34.7	196 ± 16.3	70 ± 5.4
2	60.9 ± 3.33	27.2 ± 1.14	33.7 ± 2.74	1250 ± 64.5	341 ± 10.7	139 ± 37.0
3	52.4 ± 2.37	25.5 ± 0.42	33.3 ± 5.03	1725 ± 343.1	287 ± 31.6	137 ± 12.1
4	46.8 ± 1.52	22.9 ± 0.53	24.2 ± 1.55	1462 ± 133.6	286 ± 21.7	164 ± 12.5
**Group**	**Total** **bilirubin, µmol/L**	**Conjugated** **bilirubin, µmol/L**	**Conjugated bilirubin,** **%**	**Cholesterol, mmol/L**	**Glucose, mmol/L**	**LDH,** **U/L**
1	10.9 ± 0.34	7.7 ± 0.37	71 ± 1.3	1.67 ± 0.016	3.38 ± 0.066	1974 ± 156
2	18.4 ± 0.68	17.2 ± 1.13	93 ± 2.7	3.54 ± 0.257	5.83 ± 0.561	2985 ± 345
3	19.5 ± 1.74	17.8 ± 0.42	93 ± 7.2	3.32 ± 0.332	6.30 ± 0.600	2835 ± 319
4	15.4 ± 0.67	13.5 ± 1.07	87 ± 3.3	2.83 ± 0.084	5.12 ± 0.425	2839 ± 207

## Data Availability

The source code used in this study is published in the GitHub repository https://github.com/AxelRolov/hepato-derep-lignin (accessed 15 December 2022).

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
