# Peer review of "Hepatoprotective Activity of Lignin-Derived Polyphenols Dereplicated Using High-Resolution Mass Spectrometry, In Vivo Experiments, and Deep Learning"

_ijms, 2022, doi:10.3390/ijms232416025_

Round 1

Reviewer 1 Report

The manuscript entitled: "Hepatoprotective activity of lignin-derived polyphenols dereplicated using high-resolution mass spectrometry, in vivo experiments, and deep learning" is an original paper. The authors studied the hepatoprotective activity of the lignin-derived natural compound mixture by in vivo experiments. The authors concluded that synthetic multicomponent polyphenolic agent, BP-Cx-1, showed its pronounced hepatoprotective activity in mice.

The manuscript is well structured and well written. The methods are well described and the results are clearly presented. However, introduction seems too long.

The results could have potentially implications in NAFDL treatment. Therefore the results might have practical consequences.

Author Response

The are thankful for the positive evaluation of our work and comments on our paper.

Reviewer 1.

The manuscript entitled: "Hepatoprotective activity of lignin-derived polyphenols dereplicated using high-resolution mass spectrometry, in vivo experiments, and deep learning" is an original paper. The authors studied the hepatoprotective activity of the lignin-derived natural compound mixture by in vivo experiments. The authors concluded that synthetic multicomponent polyphenolic agent, BP-Cx-1, showed its pronounced hepatoprotective activity in mice. The manuscript is well structured and well written. The methods are well described and the results are clearly presented. However, introduction seems too long. The results could have potentially implications in NAFDL treatment. Therefore the results might have practical consequences.

We are grateful for the positive feedback on our work. In fact, we believe that synthetic polyphenol mixtures may be accepted eventually as liver-supporting drug. In the revised version we have excluded excessive narrative and move a part of it to the discussion section.

Reviewer 2 Report

In this paper, Orlov et al. discuss the hepatoprotective effects of lignin-derived polyphenols with very high quality.

There is absolutely no disagreement about the research.

The introduction is very long and complex. Many sections could be moved to a new Discussion section.

In the field of nutrition, lignin as a dietary fiber may be introduced as it helps in the elimination of residues in the intestinal tract, improves the intestinal environment, and helps in weight loss, etc., and its function would be pointed out.

In line 37, it is introduced as "NAFLD may be caused by many factors including diet and adverse drug reactions," but drug-induced liver injury is not defined as NAFLD.

Also, "NADFL" is misprinted in the abstract.

Author Response

We are grateful for the positive feedback and the suggestions.

Reviewer 2.

In this paper, Orlov et al. discuss the hepatoprotective effects of lignin-derived polyphenols with very high quality. There is absolutely no disagreement about the research.

We are thankful for the positive evaluation of our work.

The introduction is very long and complex. Many sections could be moved to a new Discussion section.

Following reviewer’s suggestion we have added the discussion section to the manuscript, which also enabled to shorten the Introduction.

In the field of nutrition, lignin as a dietary fiber may be introduced as it helps in the elimination of residues in the intestinal tract, improves the intestinal environment, and helps in weight loss, etc., and its function would be pointed out.

Following reviewer’s recommendation, we have added to the Discussion the possible role of the lignin derivative as a dietary product.

Line 340: Overall, this pilot study provides a new strategy to study the mechanism of action of natural multicomponent mixtures. This requires a combination of chemical, chemoinformatic and biological experiments. In the particular case of BP-Cx-1, the results are promising from a practical point of view. In fact, synthetic lignin derivatives have batch-to-batch reproducibility[40]. With the proven mechanism it may be introduced as a bivalent drug: as a dietary fiber, which helps in the elimination of residues in the intestinal tract and improves the intestinal environment[41,42], and as a hepatoprotective agent, which prevents or reduces liver damage.

In line 37, it is introduced as "NAFLD may be caused by many factors including diet and adverse drug reactions," but drug-induced liver injury is not defined as NAFLD. Also, "NADFL" is misprinted in the abstract.

Thank you for pointing this out. We have changed the abstract and eliminated the reference to the toxic drug effect.